



**Geomorphic imprint of high mountain floods: Insight from the 2022**
**hydrological extreme across the Upper Indus terrain in NW Himalayas**
Abhishek kashyap[1], Kristen Cook[2], Mukunda Dev Behera[1*],
**[1]**Centre for Ocean, River, Atmosphere and Land Sciences (CORAL), Indian Institute of
Technology Kharagpur, Kharagpur- 721302, West Bengal, India
**[2]**ISTerre, Université Grenoble Alpes, 1381 Rue de la Piscine, 38610 Gières, France
**Emails:**
Abhishek Kashyap: kashyap95abhishek@kgpian.iitkgp.ac.in
Kristen L. Cook**:** kristen.cook@univ-grenoble-alpes.fr
(*Correspondence): Mukunda Dev Behera: mdbehera@coral.iitkgp.ac.in





## Abstract

The interaction of tectonics, surface processes, and climate extremes impacts how the landscape responds to extreme hydrological events. An anomalous precipitation event in 2022 occurred during the monsoon season along the lower reaches of the Upper Indus River, resulting in short-lived high-magnitude flooding and socioeconomic disruption downstream. To understand the spatial relationship between the geomorphic response and climatic controls of this flood event, as well as their primary triggers, we performed a landscape analysis using topographic metrics and quantified the causal association between hydro-climatic variables. Temperature anomalies in upstream glaciated sub-catchments had a considerable impact on snow cover distribution, based on our observations. As snow cover changed, glacial melt runoff rose, contributing to increased fluvial stream power after traversing higher-order reaches. The higher-order reaches of the Upper Indus River received an anomalously high amount of precipitation, which, when combined with substantial glacial and melt discharge, contributed to an extreme flood across the high-relief steep gradient channels. The flood-affected regions had a high mean basin ksn and SL-index, including numerous spikes in their magnitudes along their channel profiles downstream. To determine how the lower reaches of the Upper Indus River responded to this flood event, we employed the Enhanced Vegetation Index (EVI) and Normalized Difference Water Index (NDWI) as change indicator metrics. We observed an inverse causal influence of NDWI on EVI and a statistically significant relationship between anomalous stream power and relative EVI, suggesting that downstream channel morphology changed rapidly during this episodic event and highlighting EVI as a useful indicator of geomorphic change. We suggest that this extreme flood event is a result of the interaction of anomalous glacial melt and anomalous precipitation over a high-relief landscape, with a certain causal connection with anomalous temperature over the event duration. The synoptic observations suggest that this meteorological condition involves the interaction of the Indian Summer Monsoon (ISM) and Western Disturbance (WD) moisture fluxes. However, the geomorphic consequences of such anomalous monsoon periods, as well as their influence on long-term landscape change, are still unclear.

**Keywords:** anomalous precipitation; extreme flood; causal relationship; Upper Indus terrain



## 1. Introduction

High mountain floods in the Himalayas are associated with several processes, including coupling of the Indian Summer Monsoon (ISM) and western disturbance (WD) circulations (Houze et al., 2011), cloudbursts (Dimri et al., 2016), anomalous precipitation, cloud-scale interconnected atmospheric anomalies (Dimri et al., 2017), and geomorphic driven surface processes (Sharma et al., 2017). There is growing recognition that landscapes may evolve through the cumulative effects of extreme episodic events, in particular in rapidly eroding terrains (Korup, 2012; Cook et al., 2018). Recent studies suggest that even minor shifts in weather patterns can have a significant impact on the frequency and magnitude of floods (Knox, 2000; Liu et al., 2015; Benito et al., 2015; Sharma et al., 2022). It has also been suggested that high-magnitude flood occurrences in the bedrock rivers draining the Himalayas are the geomorphic agents with the most significant impact on the evolution of the regional landscape as well as on the residents of the downstream regions (Bookhagen et al., 2005a; Sharma et al., 2017; Panda et al., 2020).

The Tibetan Plateau and its surrounding mountainous regions, such as the Himalayas and the Karakoram ranges, are critical for the downstream hydrology and water availability of the Indus River system (Hewitt, 2009; Immerzeel et al. 2010) (Fig.1). The majority of the hydrological budget of Indus River comes from precipitation, snowmelt, and glaciers, but their relative contribution varies among the major contributing tributaries (Bookhagen and Burbank 2010; Wu et al., 2021). The Upper Indus catchment receives precipitation from two distinct climatic systems, the WD and the ISM, over its foreland and highlands in the northwest (NW) Himalayas (Bookhagen and Burbank 2006; 2010). However, it remains unclear yet how these two distinct circulation patterns interact over the Himalayan landscape and what is their potential influence on long-term geomorphic change (Dimri et al., 2015;2017; Ray et al., 2019).





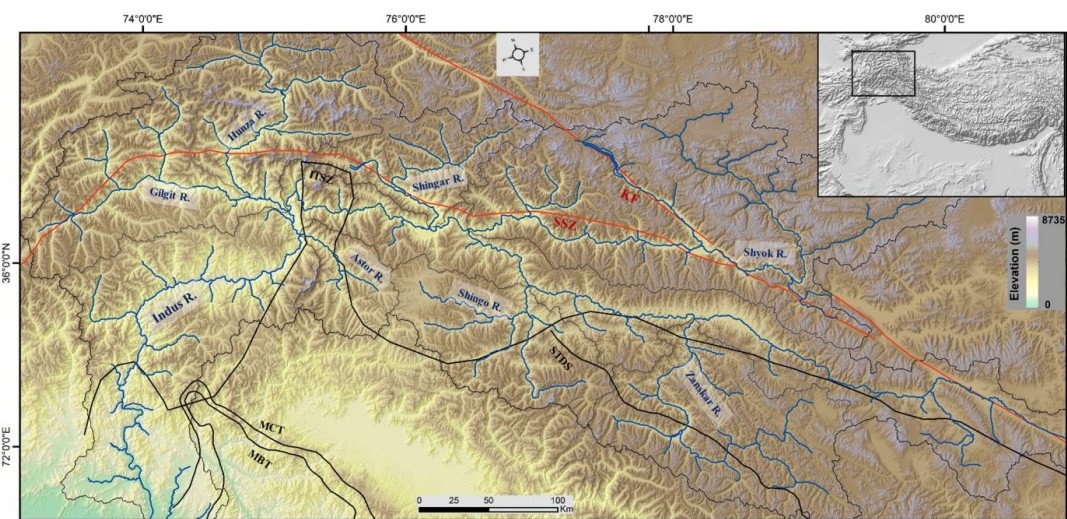

Fig.1. Regional topographic setting of Upper Indus catchment along with its major tributaries overlaid with major geological structures (MBT= Main boundary Thrust, MCT= Main Central Thrust, STDS= Southern Tibet Detachment system, ITSZ= Indus Tibetan Suture Zone, SSZ= Shyok Suture Zone, KF= Karakoram fault):

Short-duration episodic weather events have a significant influence on hillslope-surface processes and rates of bedrock erosion by modulating mass movement and subsequent landscape evolution (Snyder et al., 2003; Bookhagen et al., 2005b; Srivastava et al., 2017). During such events, a lot of sediment is transported through the fluvial system, some of which is temporarily deposited in low-gradient reaches and changes the landscape, before being finally deposited in oceanic sinks (Goodbred, 2003; Panda et al., 2020). The geomorphic signatures of catchment morphology are vital for understanding and identifying the channel response involved in such events as well as the processes and patterns of erosion (Kashyap and Behera., 2023; Sharma et al., 2017).

From the beginning of July until the end of August 2022, large portions of the Indus catchment experienced unprecedented monsoon precipitation (Otto et al., 2023; Nanditha et al., 2023). Some recent studies suggest that the primary trigger of this anomalous precipitation event was an intensely low atmospheric circulation pattern, low sea surface temperatures across the eastern Pacific, and the advent of a La-Nina event (Otto et al., 2023; Nanditha et al., 2023). This extreme precipitation event resulted in a catastrophic flood in the low elevation flood plains of the Indus catchment (Jones, 2022; Otto et al., 2023; Ma et al., 2023). This severe flood had an extreme impact over the southern province of Pakistan, causing internal displacement



of about ~30–32 million people and the deaths of ~1500–1600 people (Bhutto, 2022; Khokhar,
2022; UNICEF, 2022; Ma et al., 2023). In excess of ~\$25–30 billion in economic losses are
anticipated (Bhutto, 2022; Otto et al., 2023). According to reports, the flood in 2022 exceeded
the peak flow rate of the disastrous 2010 floods that occurred over Pakistan (Bhutto, 2022;
UNICEF, 2022; Nanditha et al., 2023). The magnitude of the fluvial discharge over the
upstream tributaries of the Indus River increased predominantly as a result of increased
streamflow across glaciated channels (NDMA, 2022; UNICEF, 2022). However, the
geomorphic consequences and the main drivers of this high-magnitude flooding in the Upper
Indus catchment have not been evaluated yet.
In the present study, we evaluated the spatial distribution of channel changes in the
mountainous portion of the Upper Indus catchment due to the extreme precipitation event in
the months of July and August 2022. We employed a channel slope-discharge product along
the trunk channel of the Upper Indus River to estimate the anomalies in the stream power
resulting from the anomalous precipitation event during July and August 2022. We used a
random-forest-based machine learning approach to compare the observed and predicted
intensity of precipitation and runoff by assessing the mean climatology of independent hydro-
climatic variables. We further quantified the causal relationship between hydro-climatic drivers
using nonlinear time series data over the event duration. We investigated the channel response
of this episodic flood event by using the NDWI and EVI as change indicator metrics and
comparing that to event characteristics such as anomalous precipitation, stream power, and
channel metrics. We want to better understand the controls on where and when these types of
extreme hydrological events will substantially modify rivers and landscapes so associated
geomorphic hazards can be better anticipated, and we also want to better constrain the potential
role of these episodic events in driving long-term geomorphic change across the western
syntaxial region.
**2. Regional Setting**
In the Himalayas, the erosion rates are high and the landscape of the mountainous terrain is
shaped by the interactions between river systems and the basement tectonics (Jaiswara et al.,
2019; 2020). Among the Himalayan River systems, the Upper Indus is unique, including a fully
developed, ~1200-1400 km long, 8th- 9th-order drainage that enters the Himalayan terrain as an
antecedent channel and cuts right over the seismically active belt in the Indus- gorges (Fig. 1).
This catchment is highly affected by recurrent landslides or debris flows, and episodic glacial





and landslide dams that represent significant geomorphic hazards (Korup & Montgomery
2008; Korup et al., 2010).

The Upper Indus River flows through the highly tectonically active region of the Nanga

Parbat-Haramosh Massif (NP-HM), which is one of the highest relief regions on earth (~>5000
m), and has the strong potential to rapidly erode uplifted material (Leland et al. 1998; Shehzad
et al. 2009; Korup et al. 2010). The NP-HM region experiences the highest recorded rates of
denudation and channel incision on earth (~12 mm/y), as well as high rates of tectonic uplift
(~4 -10 mm/y) and forms river anticlines across extremely weak crust (Koons et al., 2002;
2013; Zeitler et al., 2001; 2014; Butler, 2019). This has a significant impact on the tectonics
and morphology of the western Himalayas (Hewitt, 2009; Zeitler et al., 2014). The Upper Indus
catchment (UIC) is characterized by extremely steep channel gradient of ~>20-25º, high
topographic relief of ~4000–5000 m, and a large portion of snow covered peaks (Hewitt, 2007;
Farinotti et al., 2020).

As a fraction of the total annual discharge, snowmelt constitutes up to 50% in the

Upper-Indus catchment (UIC) (Burbank & Bookhagen, 2006; 2010; Scherler et al., 2011). Due
to the Western Disturbance (WD) inclination, the UIC has a lot of precipitation in the winter
and spring (Kapnick et al., 2014), while due to the orographic barrier of the high mountains,
the influence of the ISM in the region weakens towards to the north-west (Forsythe et al., 2017).
The annual precipitation in the UIC increases with the elevation; across the northern valley
floors- in the rain shadows it ranges from 100-200 mm/y; while at elevation ~4000-4400
ma.s.l., it ranges from 600-800 mm/y; and above >~5000 ma.s.l., it ranges from 1500 -2000
mm/y (Sharif et al., 2013; Wu et al., 2021). From October to March, the monthly mean
temperatures in the UIC are below freezing at elevations > ~3000 m (Archer, 2004). Discharge
in the tributary channels of the Upper Indus River that depend on glacier meltwater has a strong
association with summer time mean air temperatures across the Karakoram ranges (Forsythe
et al., 2017; Wu et al., 2021).

## 3. Materials and Methodology

### 3.1 Data Used

In the present study, we used a 30 m SRTM digital elevation model (DEM) for landscape
characterization and geomorphic quantative parameter estimation. In order to investigate how
the climatic variables driving this extreme event affect the processes of regional erosion, we



used 40 years' (1982–2022) duration of daily precipitation datasets from the July 1 to August
31 period from CHIRPS (Climate Hazards Group Infrared Precipitation with Station Data)
(Version 2.0 Final). Using the climatology of daily datasets from July 1 to August 31, we
observed the spatiotemporal occurrence of hydro-meteorological variables. These variables
included 2-m air temperature, skin temperature, dewpoint temperature, snowmelt, and runoff
acquired from ERA5-Land Daily Aggregated-ECMWF Climate Reanalysis with a spatial
resolution of 11132 meters. We used remote sensing-based indices, such as MODIS-derived
normalized difference water index (NDWI), normalized difference snow index (NDSI), snow
albedo, EVI, and surface reflectance bands b1 and b2, with a spatial resolution of 500 meters,
for anomalous change indicators.
**3.2 Drainage network extraction and landscape analysis.**
We extracted the drainage network from the DEM using the ArcGIS platform. A regional slope
map was produced by running a 1000 m radius mean filter over the slope model derived from
the DEM, and a regional relief map was generated by passing a 1000 m circular radius focal
range window over the DEM. Further analysis of the DEM and the derived flow accumulation
data were performed in MATLAB using the transient profiler tools (Jaiswara et al., 2019,
2020). We extracted the longitudinal profiles of the bedrock channels within an accumulation
region of about $1*10^6$ m$^2$ and channel network of the Upper Indus catchment using
TopoToolbox (Wobus et al., 2006; Kirby and Whipple, 2012; Schwanghart and Scherler,
2014). We used a 1000 m smoothing window and a 20 m vertical interval to sample the channel
networks in order to reduce the noise and artefacts that are embedded in the elevation data.
**3.3 Quantitative Geomorphic parameters**
We used geomorphic quantitative parameters such as SL (Stream length-gradient index)-index,
$k_{sn}$ (Normalized steepness index) and Stream power of the Upper Indus trunk channel to
evaluate the influence of the high magnitude flooding event across the Upper Indus River
during July and August 2022. To evaluate the spatial variability of the flood magnitude and the
channel morphology, these metrics are plotted on the longitudinal profile of the trunk channel.
**3.3.1 Stream length-gradient index (SL- Index)**
Rivers often achieve an equilibrium or steady state between erosion and sedimentation, which
is represented by a concave longitudinal river profile (Schumm et al., 2002). Tectonic,
lithological, and/or climatic factors often result in shifts in river profiles from this expected



steady-state condition (Hack, 1973; Burbank and Anderson, 2011). Here, we use the Stream
Length-Gradient (SL) index to identify the zones of topographic break and changes in the
channel gradient of the longitudinal profile by using the equation:
$$SL = (\Delta H/\Delta L)/L \ldots\ldots\ldots\ldots\ldots. (1)$$
where SL denotes the steepness or gradient of the profile for the local reach, L is the total river
length from the midpoint of the local reach to the highest point on the channel, $\Delta H$ is the change
in elevation over the reach and $\Delta L$ is the length of the reach, so $\Delta H/\Delta L$ represent the channel
slope or gradient of the reach. A sharp lithological variation and/or the differential uplift across
active structures are frequently linked to an abrupt change in SL-index along the river (Hack,
1973; Jaiswara et al., 2020; Kashyap et al., 2024).
**3.3.2 Channel Steepness index**
We extracted the bedrock profile of the Upper Indus River, which can be described using the
power law relationship between upstream drainage area (A) and channel gradient (S) as
(Jaiswara et al., 2019, 2020; Kashyap et al., 2024):
$$S = k_s A^{-\theta} \ldots\ldots\ldots\ldots (2)$$
where $k_s = (E/K)1/n$ is the channel steepness index, $\theta = (m/n)$ is the channel concavity index,
m and $n$ are positive constants, E is the erosion rate at a steady state (Wobus et al., 2006; Kirby
and Whipple, 2012).  The relative magnitude of $k_s$ is often related to the surface uplift rate as
well as the erosional efficiency across a bedrock catchment (Snyder et al., 2003; Wobus et al.,

2006).


**3.3.3 Stream Power estimation**
The normalized steepness index ($k_{sn}$) has emerged as an important topographic metric with
significant correlation with erosion rate over a wide range of timescales (Wobus et al., 2006;
Jaiswara et al., 2019; Kashyap et al., 2024). However, one major drawback of $k_{sn}$ is that it
includes an assumption of spatially constant precipitation because upstream drainage area is
used as a proxy for discharge (Adams et al., 2020; Leonard et al., 2023a).

In the present study, we incorporate the precipitation intensity into the stream power law

to analyze the anomalous stream power along the trunk channel during the flood event. We




estimate the precipitation induced stream power using the slope-discharge method, which involves multiplying the accumulated flow distance weighted by precipitation with the hyperbolic tangent function of the channel gradient along the flow path (Adams et al., 2020; Leonard et al., 2023b). The estimation of stream power ($K_{sn}Q$) as a function of channel discharge can be estimated as:

$$\mathbf{K_{sn}Q} = (S) \times f\left(\int p * FD\right)\ldots\ldots (3)$$

where S is the channel gradient, FD is the accumulated flow distance, p is the accumulated precipitation (Leonard et al., 2023a; b). Thus, $K_{sn}Q$ is a normalized version of the channel steepness metric that uses the product of channel gradient (S) and upstream discharge (Q) estimated from mean precipitation (P) as a fluvial discharge proxy. This enables $K_{sn}Q$ to account for the spatial and temporal variability in precipitation along the upper Indus River during the high magnitude flood event. Accumulated precipitation resolves spatial patterns well and scales nearly linearly with relevant discharges, particularly for large and long-lasting precipitation events (Rossi et al., 2016; Leonard et al., 2023a; b).

**3.4 Machine learning based approach to model the anomalous event characteristics**

The Random Forest (RF) technique is a supervised machine learning method that has been used as a tree-based ensemble technique and includes a bagging or boot-strapping algorithm (Breiman, 2001; Wolfensberger et al., 2021). In the present study we use a RF based multivariate regression approach to estimate the anomalous precipitation and runoff intensity in July and August 2022 using the independent variables obtained from classifying variable importance.

$$H(x) = \sum_{i=1}^{T} hi(x)\ldots\ldots\ldots\ldots (4)$$

Where, hi (x) denotes the $i^{th}$ regression tree output (hi) on sample x. Therefore, the prediction of the RF is the mean of the predicted values of all the decision trees. T denotes the regression trees for regression prediction.

Based on the mean climatology of the last 40 years, we predict the daily anomalous precipitation and runoff intensity for the 2022 event and compare them with the observed actual values. We employed the highest significance variables, as well as precipitation and runoff data from 1982 to 2021, as a training set. To utilize the independent variables to estimate these event characteristics, we first classify the hydro-climatic variables based on their higher importance





using the RF classification approach. Then, using the RF multivariate regression approach, we
select only those independent variables with the highest significance to estimate anomalous
precipitation and runoff intensity during the event duration.

**3.5 Causal discovery among Hydro-climatic variables**

Causal methodologies have been utilized to evaluate whether and how changes in one hydro-
climatological variable during anomalous extreme events influence the magnitude of another
(Runge et al., 2019a; Nowak et al., 2020). To understand how an extreme event is regulated
over high mountainous terrain, a temporal investigation of event characteristics is required.
Using this evaluation, we gain insight into how the conditioning hydro-climatic variables that
regulate these extreme events evolve throughout event duration in a catchment (Runge, 2018;
Krich et al., 2020). Understanding directional dependencies is crucial to distinguish them from
connections that cannot be deduced using any statistical model (Kretschmer et al., 2017;
Karmouche et al., 2023).
In this study, we use causal stationarity, and the absence of contemporaneous causal effects
for the time series datasets using the PCMCI and MCI approaches (Tibau et al., 2022; Runge,
2023). PCMCI is a causal identification technique that combines the Momentary Conditional
Independence (MCI) approach with the PC algorithm (Runge et al., 2019b; Nowack et al.,
2020). Given a set of multivariate time series, PCMCI estimates the time series graph that
depicts the conditional independencies among the time-lagged factors (Runge et al., 2014;
2019a). In addition to PCMCI, we use the ParCorr linear independence test based on partial
correlations is employed (Runge et al., 2014; 2023).
In the present study we use the daily datasets of hydro-climatological variables and group
them as; Temperature gradient (Tg), including Air temperature, Surface temperature, and
Dewpoint temperature; Rainfall gradients (Rg), including Precipitation intensity, Runoff and
Snowmelt; and anomalous change indicators (Ac) including EVI, NDWI, and NDSI, July 1 to
August 31, 2022, so includes 62 observational intervals. We evaluate the causal interference
between these hydro-climatic variables using the MCI approach with a maximum 2-day lag
period ($\tau_{max} = 2$) and a limit for significance set to 0.05 ($\alpha = 0.05$), in order to examine the
spatially interdependence relationships among each of these variables during 2-day event
periods.

**3.6 Moisture pathways**



The Hybrid Single-Particle Lagrangian Integrated Trajectories (HYSPLIT) model
(https://www.ready.noaa.gov/HYSPLIT_traj.php) has been employed to determine the
probable moisture parcel source region (Joshi et al., 2023). Over the past decade, researchers
have used the HYSPLIT model to identify moisture sources (Wang et al., 2017; Joshi et al.,
2023). To determine the backward trajectory following an anomalous precipitation event, this
study used the HYSPLIT model. We used three starting heights of 500, 1000, and 3000 ma. s.l
to calculate the backward trajectory for each day of July and August 2022, given that the
HYSPLIT model required the start date/time, location, and height for each precipitation event
(Wang et al., 2017; Gudipati et al., 2022). This study used meteorological data with a spatial
resolution of $1^o \times 1^o$ from the Global Data Assimilation System (NCEP-GDAS).

## 283   4. Results

### 284   4.1 Geomorphic analysis of the Upper Indus terrain

The Indus River is around ~1400–1600 km long and forms multiple loops both parallel to and
in opposition to the regional structural trend; its bed elevation ranges from ~500-6000–m. The
river exhibits distinct morphological characteristics over its flow path in terms of its
topographic attributes and derivatives. Over the elevated low-relief landscape in the Tibetan
plateau, the relief and channel gradient vary as (~0-500 m; ~0-10°), with a low SL index (~ <1
$*10^4$) gradient meter and mean basin $k_{sn}$ of (~<90 $m^{0.9}$) (Fig. 2; Fig. 3a). Then, when the river
traverses through the NP-HM region, there is a progressive rise in the local relief and channel
gradient to (~>2000-3000 m; ~>25-35°), which is also reflected in the SL-index (>2.5-3×$10^4$)
and mean $k_{sn}$ (~>331 $m^{0.9}$). This region is characterized by topographic discontinuities across
active structures, leading to high relief variation and topographic roughness.



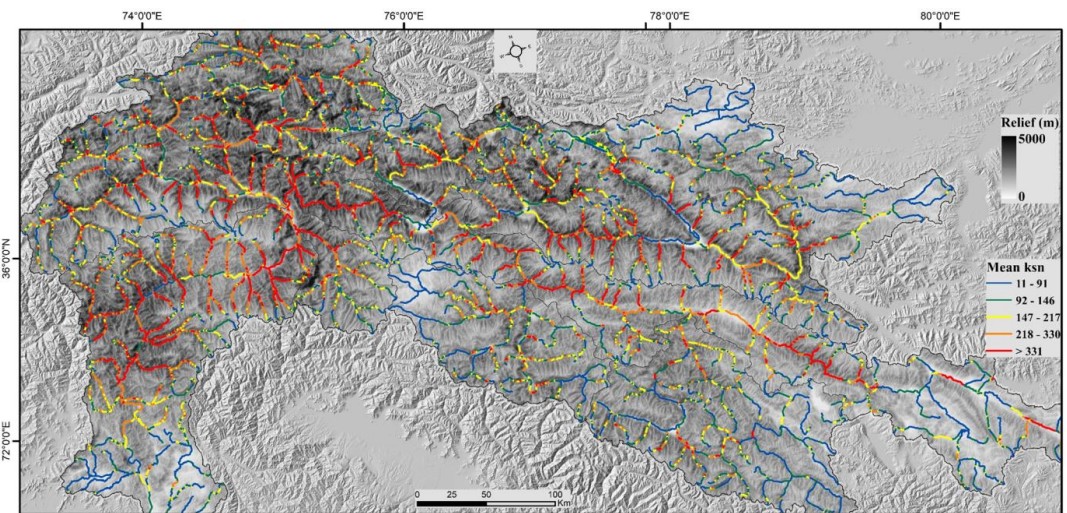


Fig.2. Spatial distribution of local relief overlaid with Mean basin ksn ranges across the Upper
Indus River catchment.

The tributaries in the upstream glaciated valleys that flow parallel to the structural trend

have a higher mean channel gradient (>~20-30º) and topographic relief (>~2000-3000 m) (Fig.
2). When these tributary channels start to descend towards the main stream after following the
glaciated landscape, the value of SL and $k_{sn}$ for the trunk channels shows a significant rise at
~3000–4000 m mean elevation. Approaching the southern mountain front, the main trunk
channel relief and channel gradient are ~1000-2000 m and ~15-25° respectively (Fig. 3a).



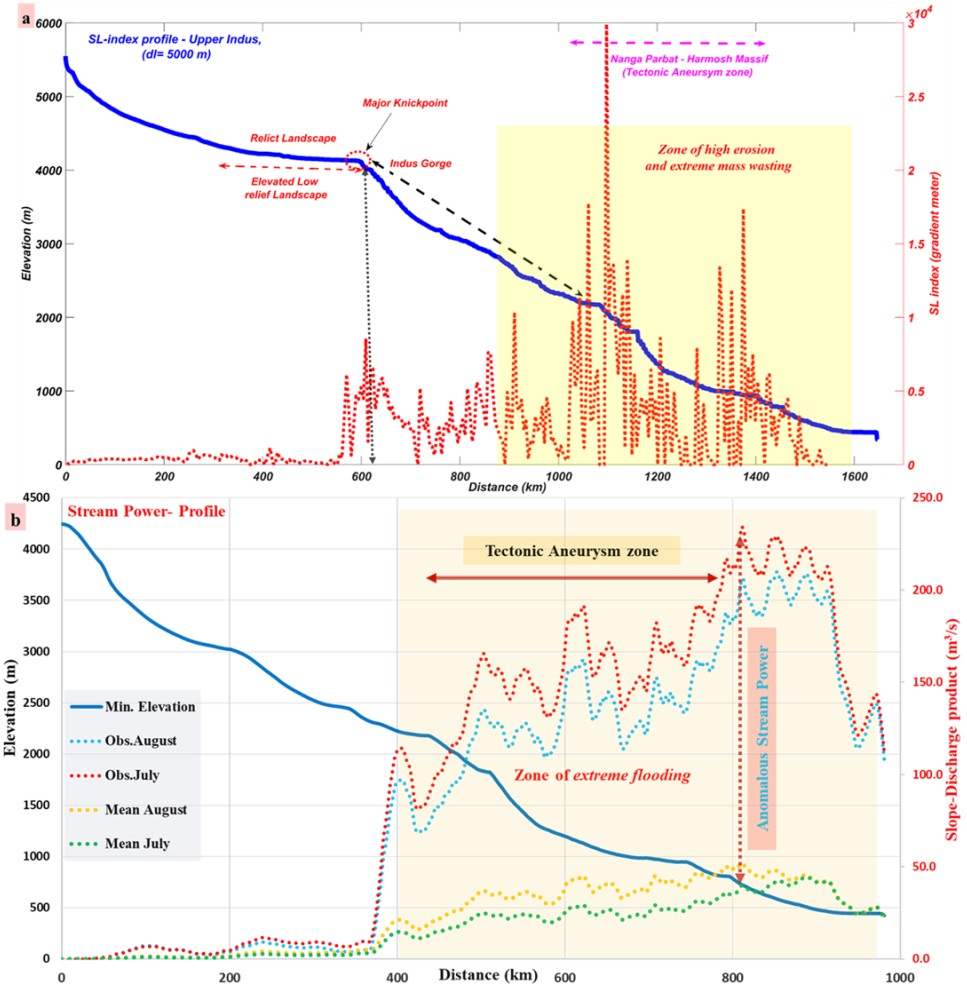

Fig.3. The trunk channel profile of Upper Indus river plotted with (a) SL-index; (b) The highest order profile of Upper Indus river plotted with Stream power (slope-discharge product)-channel elevation (highest order profile is the subset of trunk channel profile indicated by black dash line).

The spatial association of higher $k_{sn}$ (>~331 $m^{0.9}$), topographic relief (~1500-2000 m), and longitudinal increase in channel gradient along the main Upper Indus River channel downstream suggests a higher erosional regime. These high values for the various topographic metrics highlight zones of accelerated erosion where the river is in gradational disequilibrium. Furthermore, this tectonically active southern front coincides with a region that gets significant



annual mean precipitation (~1500–2500 mm/y), suggesting a tectonic-climate linkage in the
erosional process.

**4.2 Spatial distribution of Hydro-climatic anomalies over event duration**

The downstream reach of the Upper Indus trunk channel received significant amount of
anomalous precipitation (>~60–80 mm/d) during the observation period of July and August
2022 (Fig. 4a, 4b). The spatial variability of anomalous precipitation varies with a range of
>~0–40 mm/d along its major glaciated tributaries, such as Hunza, Astor, Gilgit, Shingo, and
Zanskar. In July and August 2022, the total extent of anomalous precipitation was around
~900–1000 mm/month, which was approximately ~300–400% more than the long-term (1982–
2022) mean climatology. From July to August 2022, there was continuous precipitation in the
high gradient downstream region, and due to the antecedent weather conditions, extreme
precipitation likely produced suitable conditions for high-magnitude flooding. The potential
geomorphic response of such anomalous precipitation is suggested by the resulting anomalous
stream power over the downstream channels (Fig. 4c, 4d). The spatial distribution of
anomalous stream power shows that the greatest increase occurred at ~400-800 km along the
channel profile downstream. For both the months of July and August of 2022, we observed a
significant rise in the stream power, to ~>200 m$^3$/s above the mean values (Fig. 3b).



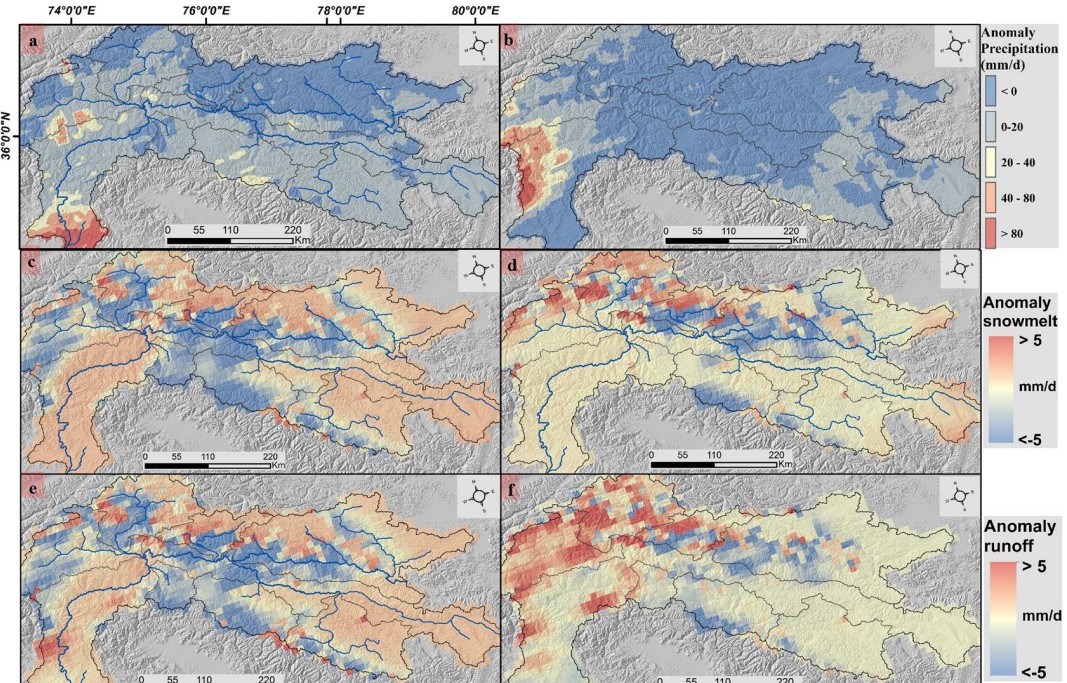

Fig.4 Spatial distribution of hydro-meteorological variables for anomalous July and August month of 2022 across Upper Indus catchment such as: (a) precipitation (July) (b) precipitation (August) (c) Snowmelt (July) (d) snowmelt (August); (e) Runoff (July) (f) Runoff (August).

During the observation period, other variables, such as runoff and snowmelt, also showed positive anomalies across the upstream glaciated sub-catchments over the Karakoram ranges (Fig. 4e, 4f). Furthermore, during July and August 2022, the temperature variables indicated a positive deviation from the mean climatological trend over the glaciated catchments. In the upstream sub-catchments in Shyok, Shingar, Hunza, and Gilgit, air and dewpoint temperatures reach (>~3°C above mean), while surface temperatures reach (>~6°C above mean) (Fig. 5). The spatial distribution of anomalous temperatures corresponds well with the anomalous snowmelt and runoff magnitude across the upstream glaciated catchments.



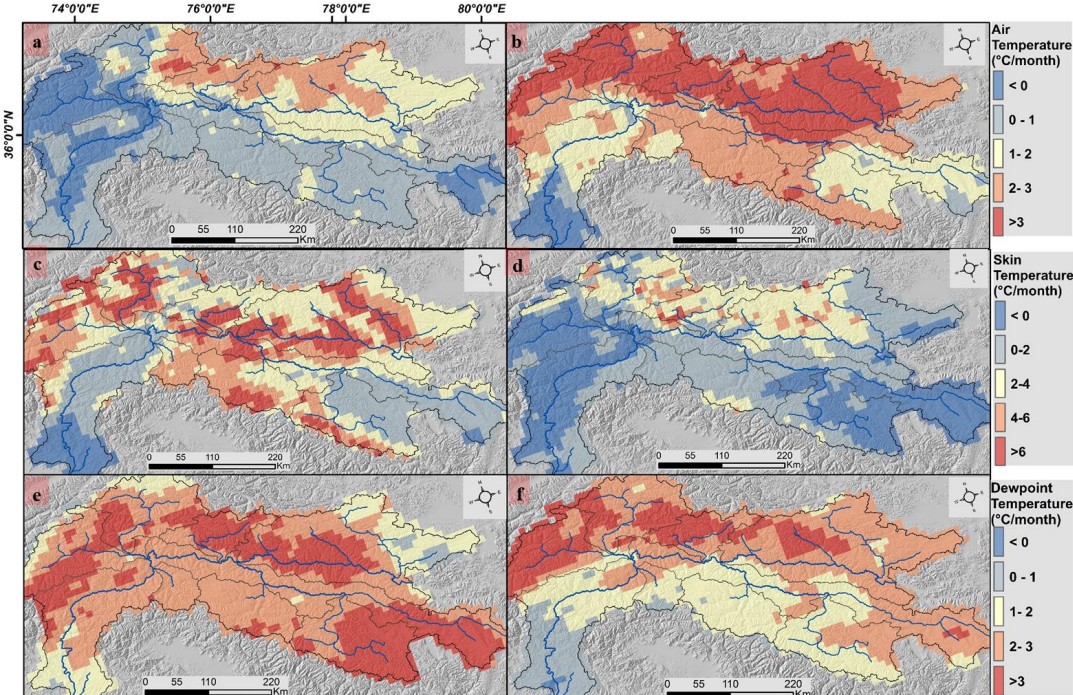

Fig.5 Spatial distribution of hydro-meteorological variables for anomalous July and August month of 2022 across Upper Indus catchment such as: (a) Air temperature (July) (b) Air temperature (August) (c) Surface temperature (July) (d) Surface temperature (August); (e) Dewpoint temperature (July) (f) Dewpoint temperature (August).

We also observed a significant shift in the spatial distribution of change indicator variables during the observation period. In July 2022, the lower reaches of the Upper Indus River exhibited a negative change in EVI (~0-0.21) and a positive relative NDWI (~0.15-0.20). This inverse relationship between these two change indicators was found in the upstream channel as well in August. During the event, the tributary channels in the upstream glaciated landscape experienced a significant change in snow cover distribution, as demonstrated by the spatial variations of the relative NDSI (~ 0-0.63). Changes in relative snow cover correspond directly to increases in snowmelt and glacial runoff across glaciated catchments (Fig. 6). We observed a significant relationship (p<0.005; R=0.81) between the relative EVI metric and the anomalous stream power in the Upper Indus trunk channel and along its main tributaries. The anomalous stream power of the Upper Indus River and all of its major tributaries corresponds to a proportion of EVI change that exceeds across low-gradient regions. This positive



relationship with an increasing trend suggests a substantial geomorphic response due to
extreme flooding. However, a negative relationship between anomalous stream power and EVI
can also be observed across the channels of Astor and Shingo (Fig. 7).

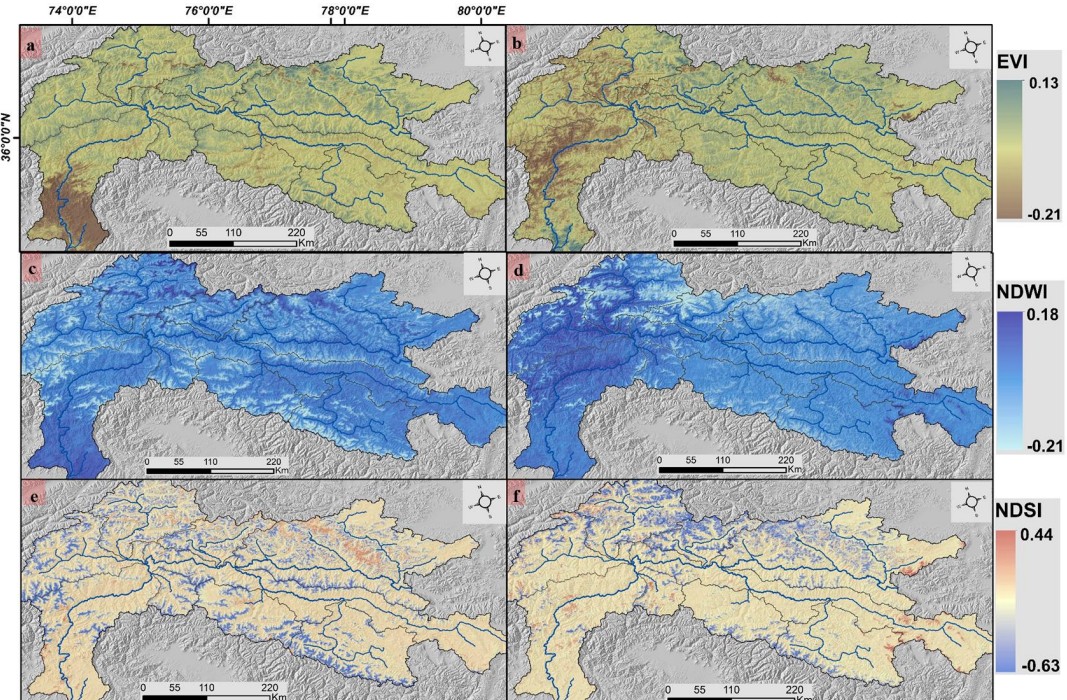


Fig. 6. Spatial distribution of hydro-meteorological variables for anomalous July and August
month of 2022 across Upper Indus catchment such as: (a) EVI (July) (b) EVI (August) (c)
NDWI (July) (d) NDWI (August); (e) NDSI (July) (f) NDSI (August).






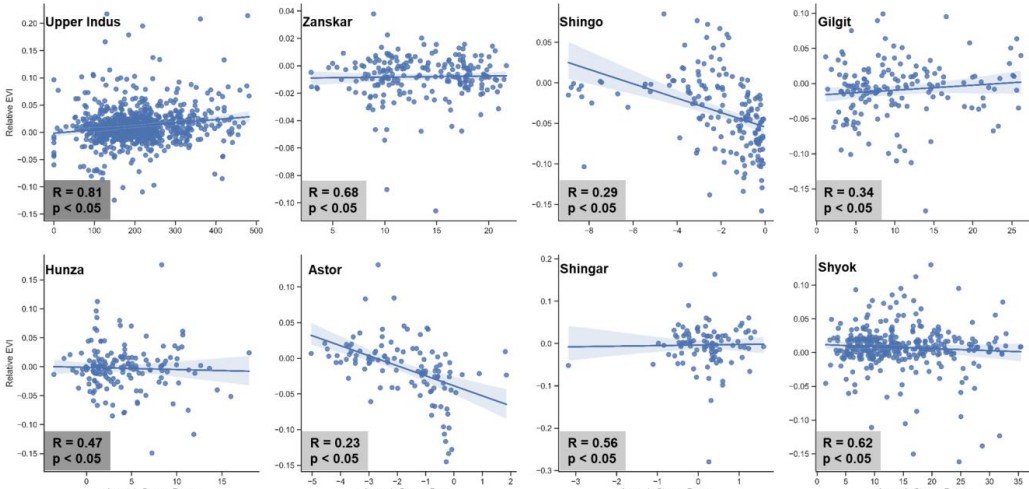

Fig.7. Statistical relationship between Relative EVI- Anomalous Stream Power from July 1 to
August 31, 2022 across Upper Indus catchment as well as along its all the major tributaries:
**4.3 Machine learning based approach to quantify the event anomalies**
The RF-classification-based determination of variable importance indicates that dewpoint
temperature is the most significant variable in estimating precipitation intensity. Other
significant variables include surface temperature and air temperature. Relative NDSI was the
variable of highest significance for estimating precipitation in all other sub-catchments except
Shingar (Fig. S1). Snowmelt, dewpoint temperature, relative NDSI, and surface temperature
are the most significant variables for each sub-catchment when estimating runoff intensity.
Surface temperature holds higher significance in the trunk channel catchment of the Upper
Indus, followed by air temperature and precipitation intensity (Fig. S2). The anomalous
precipitation and runoff intensity are then estimated using these independent variables with the
highest significance obtained during classification.
The results show that the Upper Indus catchment received significantly more
precipitation and runoff than predicted at multiple instances in July and August of 2022 (Fig.
8). The anomalous and extreme characteristics of the hydro-climatic and terrestrial drivers
could explain this phenomenon. The Upper Indus catchment received a significant amount of
anomalous precipitation, with an intensity of $>\sim100$ mm/d, which is much higher than the
predicted intensity during the period of observation. The channels in the higher relief
landscapes such as Astor and Gilgit encountered the second-highest anomalous incidence, with



intensities ~80–100 mm/d. The upstream glaciated catchments, such as the Shyok, Shingo, and
Hunza, also have persistent anomalous intensities of up to ~100 mm/d. The least impacted
catchment was Zanskar and Shingo, despite a high rate of precipitation that ranges from ~60–
80 mm/d.

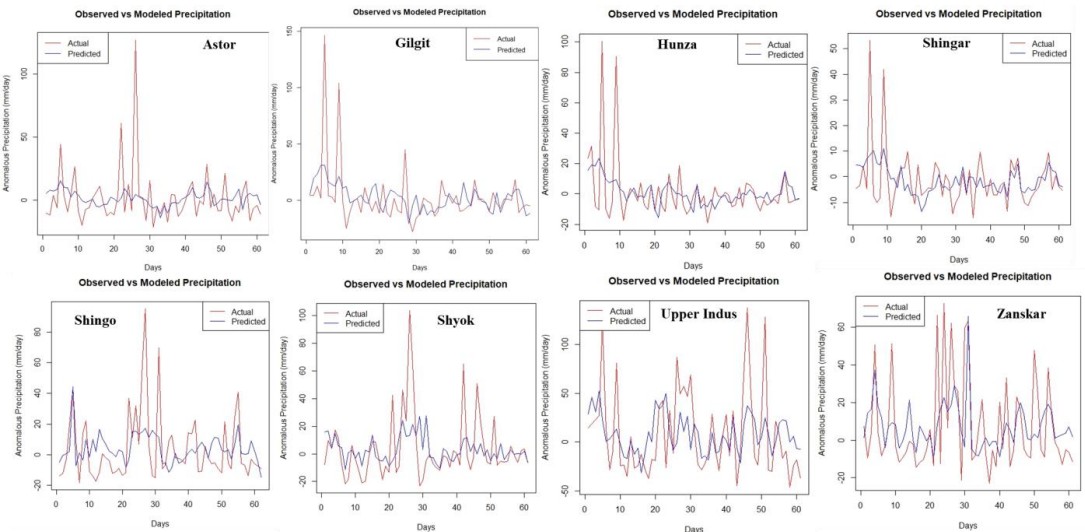


Fig.8 Random Forest-Regression based observed vs modeled anomalous precipitation from
July 1 to August 31, 2022 across Upper Indus catchment as well as along all the major
tributaries.
The distribution of observed and predicted runoff shows the intensity of observed runoff
corresponds with the precipitation trend. During the observation period, the Upper Indus
catchment had much higher runoff rates, followed by upstream glaciated sub-catchments
including Shyok (~30-60 mm/d), Shingo, and Gilgit (~20-30 mm/d). However, in the majority
of the upstream sub-catchments, the observed anomalous runoff intensity is insignificant (Fig.

9).



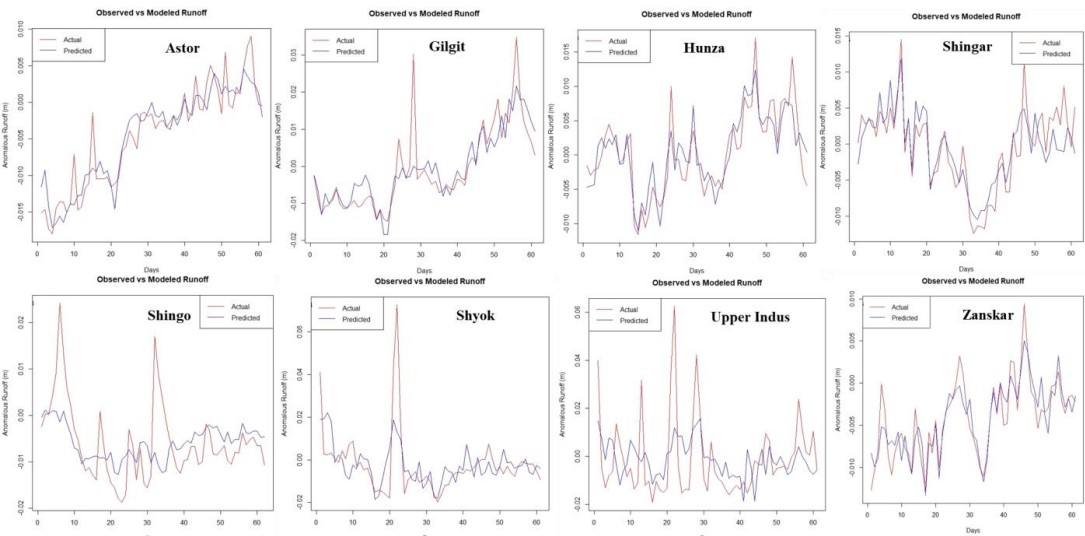

Fig.9 Random Forest-Regression based observed vs modeled anomalous runoff from July 1 to August 31, 2022 across Upper Indus catchment as well as along all the major tributaries:

## 4.4 Causal relationship among Hydro-climatic variables over event duration

The causal analysis showed that the impact of numerous meteorological variables on the extreme flood over the Upper Indus terrain varied significantly. We observed a significant causal lagged connection between dewpoint temperature and NDSI, which together positively influenced precipitation intensity with a 1-day lag across the Upper Indus catchment. Similarly, precipitation intensity and snowmelt exhibit a positive causal influence on NDWI with a 1-day lag period. For instance, the cross-correlation between precipitation and dewpoint temperature with positive impact is > 0.4 over the event duration. There was a significant negative causal influence of NDWI on EVI, indicating an inversely proportional relationship across the observational lag period. The hydro-climatic variables such as precipitation intensity, snowmelt, NDWI, EVI, NDSI, air temperature, and surface temperature, had non-linear and non-stationary tends from July 1, 2022, to August 31, 2022, as shown by the autocorrelation and PCMCI magnitude over the time series. The auto MCI ranges of these variables are comparatively very low. Runoff and dewpoint temperatures exhibit stationarity and a linear trend over the time series with relative high auto-MCI ranges. It is also observed that dewpoint temperature has a significant inherent connection with snowmelt and NDSI, indicating that these variables have a direct causative relationship with a high cross-MCI range (Fig. 10). In a causal investigation, edges with arrows indicate a link between the drivers. However,



depending on the available metrics, there may be an instant causal connection between the
drivers. It should be observed that this relationship may not have been determined to be
causative.

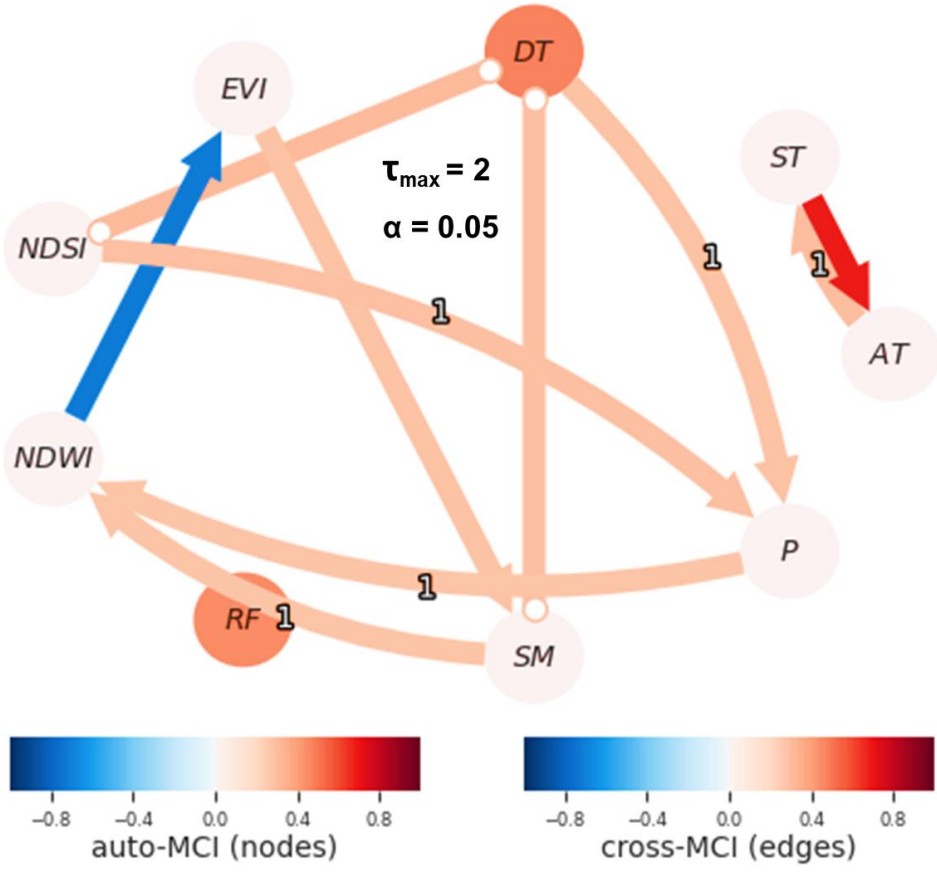


Fig.10. Causal detection among hydro-climatic driver having non-linear time series from July
1 to August 31, 2022 across Upper Indus catchment with maximum allowable lag of 2 days at
the 95% CI. (The drivers are shown in the solid circles such as: DT= Dewpoint Temperature,
ST= Surface Temperature, AT= Air Temperature, P= Precipitation intensity, SM= Snowmelt,
RF= Runoff, NDWI= Normalized Difference Water Index, NDSI= Normalized Difference
Snow Index, EVI= Enhanced Vegetation Index: The node colour represents autocorrelation
whereas link colour represents the strength of directional link. The lag at which the link was
found significant is shown as link label, absence of which indicates that the link was found at
zero lag).



### 4.5 Identifying moisture trajectories for the anomalous precipitation event

Based on moisture source uptake along trajectories for the observation period of July 1 to August 31, 2022, the amount of precipitation across the orographic ridges of the Upper Indus terrain was delivered along two major pathways, one from Mediterranean Sea sources such as Western disturbance (WD)-derived moisture during the onset of the monsoon and a second from the ISM, originating from the Bay of Bengal and the Arabian Sea. The WD routes provided the moisture sources for the precipitation along the 3000 m height trajectories, while the Arabian Sea, the Bay of Bengal, and the Himalayan foreland provided the moisture along the 500 m and 1000 m trajectories. Furthermore, the anomalous temperature gradient observed for the months of July and August 2022 shows that the steep bedrock valleys are causing abnormal air-mass feedback. The substantial divergence in the air-mass curve from mid-July to mid-August 2022 suggests there may have been very high precipitation and temperature fluctuations during those periods (Fig. 11).

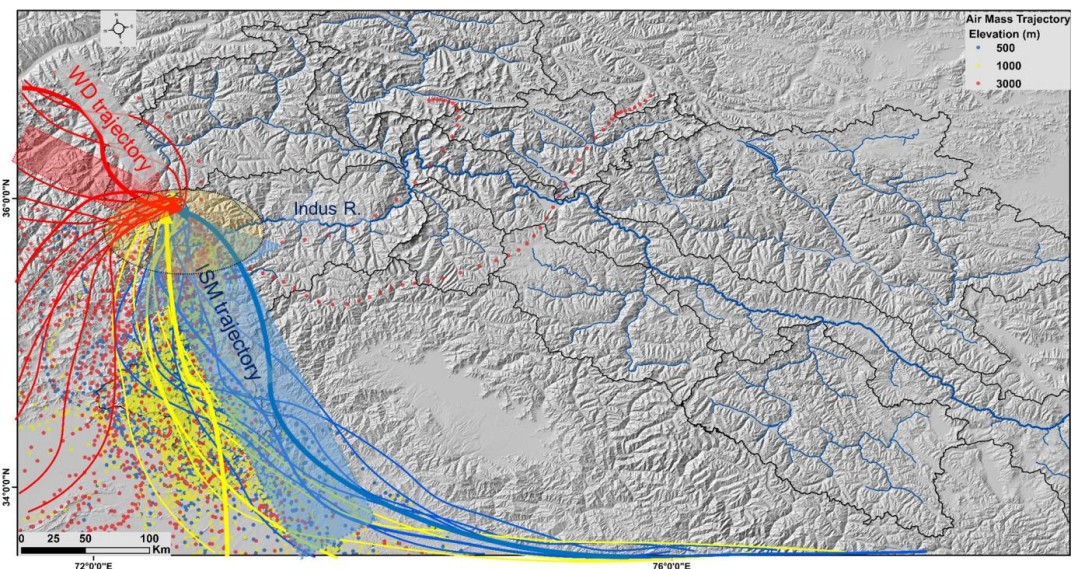

Fig.11. Moisture pathways (Backward trajectories) for Anomalous precipitation event from July 1 to August 31, 2022 across Upper Indus catchment: (Blue line denotes the trajectory of 500 m elevation, Yellow line denotes the trajectory of 1000 m elevation, and Red line denotes the trajectory of 3000 m elevation: Blue and yellow dot lines exhibits the ISM pathways, whereas Red dot lines exhibit the WD pathways).





## 5. Discussion

### 5.1 Spatial relationship between topographic metrics and event anomalies

To characterize the geomorphic response of this extreme flood, we estimated stream power over the trunk channel of the upper Indus River as an event anomaly. Understanding the spatial distribution of stream power over the longitudinal profile of bedrock rivers is essential for evaluating the catchment-scale variability in channel response to anomalous precipitation events (Whipple et al., 2000; Kaushal et al., 2020). The peaks and troughs in the stream power profile regulate the morphological characteristics of the bedrock channels (Schneider et al., 2014; Bawa et al., 2014; Sinha et al., 2017). The river morphology and channel shape will be significantly impacted by the temporal variations in flooding intensity during anomalous precipitation events (Bookhagen and Strecker, 2012; Scherler et al., 2014).

The initial ~400–600 km length of the Upper Indus River is characterized by low gradient channels as the river traverses over the elevated-low relief landscape. After traversing through the mainstream and joining in the highest-order channel across the syntaxial region, there is a sharp rise in the stream power profile along the downstream. The western syntax (NP-HM) in the NW Himalayas is one of the most rapidly uplifting (>~5-10 mm/y) and eroding (>~10 mm/y) regions on earth, with extreme topographic relief (>3000 m) (Fig. 1; 2). The sudden increase in the stream power of the Upper Indus River after traversing through NP-HM and the resultant extreme flood along lower reaches were also attributed to this high elevation change (>~4000 m) and steep channel gradient (>~20-30º) (Fig. 3b). The spatial variability of stream power is also highly connected with other topographic metrics such as the $k_{sn}$ and SL index, which demonstrate a considerable rise in their longitudinal profiles when the channel crosses the NP-HM region (Fig. 3a). We observed that the stream power distribution along the longitudinal profiles of the Upper Indus River is characterized by numerous peaks for both anomalous precipitation months in July and August 2022 (Fig. 3b).

The upstream glaciated channels of the Trans Himalayan and Karakoram ranges have a substantial glacial influence on erosion, contributing to the main trunk channel of the Upper Indus River. Therefore, such high-magnitude floods ought to propagate through the channels of high mountainous tributaries like Shyok, Gilgit, and Hunza, depending on the landscape characteristics of the upper Indus catchment. A moderate change in the distribution pattern of snow cover may have a significant impact on glacial runoff and substantially contribute to fluvial discharge. In addition to the southern mountain front, the headwaters and syntaxial zone





of the Upper Indus catchment received a significant amount of precipitation, which contributed
to the anomalous rise in stream power and substantially contributed to this extreme flood that
influenced the channel geometry of the lower reach and drove high bedrock erosion (Fig. 4).
However, the lower reaches with higher stream power are distinguished by the steep channel
valley and absence of sediment deposition. The observation suggests that the higher-order
channels of the Upper Indus River traversing across higher relief and steep gradient valleys
likely possess direct first-order control over the pattern of erosion when combined with an
anomalous rate of precipitation (Fig. 3b).
**5.2 Hydrological extremes and causal connections**
Our observations suggest that the interaction of glacial runoff with fluvial discharge over the
steep gradient channels combined to drive the extreme flood event across the Upper Indus
catchment. These extreme hydrological episodes imply that the possible response of
atmospheric instabilities may be elevation-dependent (Dimri et al., 2015; Forsythe et al., 2017;
Ullah et al., 2021; Sharma et al., 2021). It commenced with anomalous rises in temperature
gradients over the glaciated sub-catchments of the Upper Indus terrain, which drove the rapid
changes in snow cover distribution (Fig. 5; 6). This directly impacts glacial runoff magnitude
and contributes to an anomalous rise in fluvial stream power when traversed downstream over
higher-relief fluvial reaches (Fig. 6). The lower reaches of the Upper Indus catchment
witnessed an anomalous amount of precipitation intensity from early July to late August 2022
(Fig. 4).
When compared to the annual mean climatology, the precipitation intensity in the lower
reaches of the Upper Indus River was roughly ~150–200% higher in the 2022 monsoon period.
The moisture flux trajectories observed during the 2022 monsoonal period across the lower
reaches of the upper Indus River reveal two distinct sources of moisture pathways, indicating
that the combined effect of the westerlies-driven precipitation and the active monsoon phase
has likely caused this episodic event (Wang et al., 2017) (Fig. 11). Over the past years, the
interactions between moisture-laden ISM and southward-penetrating upper-level WD
depression have been linked to some catastrophic western Himalayan floods, such as in 2010
across Pakistan and 2013 in Uttarakhand, India (Rasmussen and Houze, 2012; Vellore et al.,
2015; Dimri et al., 2016; Sharma et al., 2017). This anomalous rise in the rate of precipitation
intensity contributes to the rapid increase in stream power across steep valleys. The combined
causal influence of temperature and precipitation intensity with topography plays an important



role in modulating such episodic events, as these variables eventually regulate the amount of
solid precipitation, influence the change in snow cover, and have a significant impact on
snowmelt runoff (Fig. 10) (Bovy et al., 2016; Godard and Tucker, 2021; Delaney et al., 2023).
This flood indicates the importance of understanding the cause-and-effect relationship between
temperature and precipitation in high-elevation uplands.

**5.3 Channel Response to an Extreme Flood**

We utilize NDWI and EVI as change indicator metrics to understand the changes in channel
morphology due to this extreme flood event. The spatial variability of EVI corresponds
significantly with an increase in NDWI intensity downstream during July and August 2022
(Fig. 6). This is because increasing precipitation serves a vital role in regulating relative change
in the EVI by decreasing surface albedo and temperature (Anderson and Goulden, 2011). In
some cases, correlation is due to directing flooding in vegetated areas.
The substantial decrease in EVI values along downstream channels has also been attributed
to the anomalous precipitation event, which led to increased surface runoff, higher NDWI
limits, and subsequent flood deposits. We observed a significant direct causal influence with
one-day-lagged connection of precipitation and snowmelt on NDWI (Fig. 10). This combined
causal relationship between precipitation and snowmelt with NDWI intensity indicates that
anomalous runoff occurred across both glacial and fluvial channels. Further the inverse causal
connection (negative MCI ranges) between NDWI and EVI illustrates the rapid change in the
channel geometry due to increase in the fluvial discharge over lower reaches (Fig. 10).
The change in river morphology driven by the high-magnitude flood episodes is also
documented by the statistically significant (p<0.005; R= 0.81) relationship observed between
anomalous stream power and relative EVI across the lower reaches of the Upper Indus River
(Fig. 7). It is generally assumed that relative vegetation intensity is an indicator of geomorphic
change that results from short-duration, high-magnitude hydrological events (Olen et al., 2016;
Starke et al., 2020; Clift and Jonell, 2021; Scheip and Wegmann, 2021). Thus, we anticipate
that EVI acts as a spatial indicator of change in the channel morphology across the lower
reaches of the trunk channel during the monsoon period of 2022 (Fig. 7) suggesting that the
distribution of event characteristics such as NDWI and EVI can be useful to detect the relative
change in channel morphology triggered by high-magnitude floods.



## 6. Conclusion


Our study reveals several significant event characteristics of the 2022 Upper Indus flood. Our
analysis shows that the Upper Indus flood originated across elevated glacial channels due to
the anomalous temperature rise, which increased the glacial runoff. This increase in runoff
across glaciated catchments after traversing through fluvial reaches enhanced the fluvial
discharge and likely increased the stream power in the anomalous precipitation region. The
synoptic observation of moisture pathways indicates that this anomalous precipitation incident
is linked to the interaction of southward moving mid-latitude westerlies troughs and eastward
advancing southwestern monsoon circulation. We observe a statistically significant
relationship between the anomalous stream power and relative EVI change across the lower
reaches, which serves as a significant geomorphic indicator of change in the channel
morphology. This will aid in determining the reliability of EVI as a consistent indicator of
geomorphic changes, as well as its applicability in studying the geomorphic evolution of
regional landscapes. This extreme flood illustrates how causal connections between
temperature and precipitation across high relief-gradient channels can magnify the impacts.
Such hydrological events may play significant roles as efficient geomorphic agents of erosion
and, therefore, in the coupling of climatic extremes, topography, and erosion. This study
underscores the susceptibility of the elevated syntaxial region to short-lived, high-magnitude
flooding, indicating the need for additional research to determine the causal relationship
between the drivers of hydrological extremes. Significant research is needed to understand the
long-term impact of these extreme climatic events on the geomorphic processes in the region.










**Code and data availability:**

The Data used and methodology section includes all of the open-source datasets and tools used in the study.

**Author contribution:**

Abhishek kashyap (AK): Conceptualization, Formal analysis, Methodology, writing – original draft, Writing – review & editing.

Kristen L. Cook (KLC): Supervision, Visualization, Writing – review & editing

Mukunda Dev Behera (MDB): Supervision, Validation.

**\*Competing interests**

The authors declare that they have no known competing financial interests or personal relationships that could have appeared to influence the work reported in this paper. We wish to confirm that there are no known conflicts of interest associated with this publication and there has been no significant financial support for this work that could have influenced its outcome.

**Acknowledgments:** The authors acknowledge the authorities of IIT Kharagpur for facilitating the study. AK thanks the Ministry of Education, Government of India, for the grant of a Ph.D. Research Fellowship. AK thanks the IRD "South North Scheme" scholarship, managed by Campus France, for the mobility and facilitation of a major part of this study at ISTerre, Université Grenoble Alpes.



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

58.