# Peer review of "Geomorphic imprint of high mountain floods: Insight from the 2022"

_EGUsphere, 2024_

## Author Response (AR1)

Thank you for these detailed comments and valuable feedback for this manuscript. We appreciate the comments raised by both of the reviewers, and we will address each of them individually in our response section below. We look forward to providing a revised manuscript.

**RC1: 'Comment on egusphere-2024-1618', Anonymous Referee #1**

This paper presents a valuable case study of a fascinating high-magnitude event, namely the anomalous precipitation event during the 2022 monsoon that led to substantial flooding across the lower reaches of the upper Indus River. The authors provide insight into the impacts of the flood on this high mountain region by evaluating a series of topographic indices, alongside precipitation datasets, to derive stream power proxies mapped in relation to various attributes of the flood. They find that the high precipitation volumes delivered during the event, alongside unusually high temperatures, led to substantial snow and glacier melt, contributing additional flow to the runoff generated by the high precipitation. As a result, very high stream powers were generated at multiple steep locales along the stream network, and these are assessed in relation to measures of channel change inferred from satellite-derived vegetation metrics.

The paper focuses on highlighting the combination of factors that led to the high (inferred) stream powers and therefore its main utility lies in the recognition of flood generation processes in this environment. This is important and understandably, therefore, the paper pays less attention to how the generated stream powers may (or may not) correlate with channel changes inferred from the satellite images. That is, the linkages between the stream powers generated and the channel responses are a story left largely to another analysis. The paper nevertheless represents an important contribution to our knowledge of high mountain flood generation processes and will be of interest to the journal's readership. Of particular interest is the authors' finding that for this event the atmospheric instabilities responsible for delivering the extreme precipitation and temperature anomalies were elevation-dependent, which has important implications for evaluating the controls on, and risks posed by, similar events in the future.

The paper is well-illustrated and clearly written, and I have few suggestions for further revisions. Nevertheless, the authors may wish to consider the following points:

**Author Response:** Thank you for your positive comments about our manuscript. We are pleased that you found this study clear and valuable.

**Comment 1.** The authors contextualise the flood event, but at L86 they discuss the flooding in the *Lower* Indus valley, seemingly attributing the flooding there to extreme rainfall. This is perhaps a contested view - in the lower basin could much of the flooding have been the result of the poorly maintained canal network - were flood discharges (at peak) generated in the *lower* basin unusually high or not?

**Author Response:** This is a good point. In this study, we are not looking at the portion of the Indus downstream of the mountain front, so our use of the term Lower Indus is indeed incorrect. **To make this clear, we have amended the manuscript to replace the Lower Indus with the Lower Middle Indus.**

**Comment 2.** The authors rely on the use of the CHIRPS precipitation dataset. Readers may value an assessment of any evidence that could support the reliability of CHIRPS in the study region.

**Authors Response:** Thanks for the suggestion. Various researchers extensively evaluated CHIRPS precipitation datasets at daily, monthly, and annual temporal scales across the Indus Basin. They usually divide the whole Indus basin into three sub-basins, i.e., the Upper Indus basin (UIB), the Middle Indus basin (MIB), and the Lower Indus basin (LIB). The performance of CHIRPS was comprehensively evaluated against regional ground datasets obtained from meteorological departments. Several studies (Katsanos et al., 2016, Paredes-Trejo et al., 2017, Bai et al., 2018, Gao et al., 2018, Saeidiz et al., 2018) have recommended CHIRPS for hydrological analysis and water resource management due to its fine spatiotemporal resolution. Shahid et al. (2021) conducted a study where they observed the hydrological utility of CHIRPS in two sub-basins of the Indus Basin, specifically the Gilgit (UIB) and Soan (MIB) basins. The SWAT model was used to assess the performance of CHIRPS in simulating daily streamflow across the Gilgit and Soan basins, which have completely different climates. Different regions around the world have evaluated CHIRPS (Dinku et al., 2018, Gebrechorkos et al., 2018, Prakash, 2019, Wu et al., 2019), particularly across Pakistan (Ullah et al., 2019, Nawaz et al., 2021). **We added this information to the revised manuscript.**

**Please refer to Page No: 6-7 & Line No: 150-160**

"To investigate the impact of the climatic variables driving this extreme event on regional erosion processes, we utilized daily precipitation datasets spanning 40 years (1982–2022) from July 1 to August 31 from CHIRPS (Climate Hazards Group Infrared Precipitation with Station Data) (Version 2.0 Final). Previous Several studies have investigated CHIRPS precipitation datasets at daily, monthly, and annual temporal scales across the Indus Basin (Gao et al., 2018; Ullah et al., 2019; Nawaz et al., 2021; Shahid et al., 2021). In their studies, they extensively evaluated CHIRPS's performance against regional ground datasets obtained from meteorological stations. Several studies (Katsanos et al., 2016, Paredes-Trejo et al., 2017, Bai et al., 2018, Gao et al., 2018, Saeidiz et al., 2018) suggest CHIRPS for hydrological analysis and water resource management due to its fine spatiotemporal resolution".

**Comment 3.** The authors may wish to comment in more detail on the results of Figure 8, where it is clear that the model precipitation data do not capture the full variability present in the 'observed' data. What is it about the model behavior that means the model seemingly does not represent the high precipitation variability? (Likewise for Figure 9 for the runoff data)

**Author's Response:** It appears that we may not have communicated this clearly. The model, in fact, does not accurately predict the extreme precipitation ranges of the 2022 July–August events. This is actually the aim of this machine learning-based model: to demonstrate that this event is anomalous and driven by the interaction of different circulation patterns, such as southward moving mid-latitude westerlies troughs and eastward advancing southwestern monsoons, as well as anomalous meteorological conditions, rather than the typical monsoon rainfall. **We have clarified this in our discussion of the model results in the revised manuscript.**

**Please refer to Page No: 25-26 & Line No: 544-559**

"When compared to the annual mean climatology, the precipitation intensity in the lower middle reaches of the Upper Indus River was roughly ~150–200% higher in the 2022 monsoon period. The 2022 Upper Indus flood represents an abrupt change from the region's prior precipitation and runoff patterns. To study this anomaly, we utilized a Random Forest model trained on climatological data from the last 40 years (1982-2021), with an emphasis on the months of July

and August. The model used previous climatology as a training dataset to estimate precipitation and runoff, which are significant drivers of flooding. Despite the Random Forest model's resilience, the results revealed a substantial difference between the model's predictions and the actual observed data obtained from the 2022 flood event. The model, based on 40 years of past data, failed to capture the high precipitation and runoff patterns observed in July and August 2022 (Fig. 8: 9). The model's inability to predict rainfall intensity, as well as subsequent runoff, highlights the anomalous nature of the event. This disparity demonstrates that the 2022 flood was not only unusual but also went outside the typical climatological shifts observed over the previous four decades. This emphasizes the necessity for future modeling efforts to include other predictors, such as changes in snowmelt dynamics, atmospheric circulation anomalies, and other non-stationary phenomena".

**RC2: 'Comment on egusphere-2024-1618', Anonymous Referee #2**

The manuscript "Geomorphic imprint of high mountain floods: Insight from the 2022 hydrological extreme across the Upper Indus terrain in NW Himalayas" addresses an important and timely topic in the field of fluvial geomorphology. The authors' attempt to analyze the impacts of an extreme flood event in a complex mountainous terrain is commendable, and their multi-faceted approach, combining geomorphic analysis, remote sensing, and advanced statistical techniques, demonstrates ambition and creativity in tackling this challenging subject. The study's strengths lie in its comprehensive consideration of multiple factors influencing flood response, including precipitation, temperature, snowmelt, and pre-existing landscape characteristics. The authors' use of various data sources and their attempt to link large-scale climatic drivers to local geomorphic changes is noteworthy. However, despite these positive aspects, the manuscript suffers from several critical methodological flaws that significantly undermine the validity and reliability of its findings. These issues span multiple aspects of the study, including data resolution and quality, analytical techniques, and interpretation of results. The lack of adequate pre- and post-flood comparisons, insufficient validation of remotely sensed data, problematic application of causal analysis, and inadequate error analysis and uncertainty quantification are particularly concerning. Given the severity and pervasiveness of these methodological shortcomings, I regrettably must recommend the rejection of this manuscript. The following detailed comments outline the specific issues that led to this decision, along with suggestions for how the authors might address these problems in future work.

**Author Response:** Thank you for your extensive feedback. You made several points identifying things that are missing in the manuscript, particularly related to error analysis and method robustness, and we appreciate this in helping to improve our manuscript. Several of the points raised highlight that we are not clear enough about the goals of the study, which is very helpful to know and has been improved.

**Comment 1.** Inadequate data resolution and quality: The authors rely heavily on 30m SRTM DEM data (Line 149) for their geomorphic analysis. This resolution is insufficient for accurately capturing the fine-scale topographic changes expected from a single flood event in a complex mountainous terrain. High-resolution LiDAR or drone-derived DEMs (sub-meter resolution)

would be necessary for this type of analysis. The authors mention using several datasets (Lines 148-160) but lack specificity about which data is used for each geomorphic index, and how these data were applied. At what resolution was all the data transformed for each index?

**Author's Response:** It is correct that the topographic data (DEM) that we have used is not suitable for detecting detailed geomorphic changes. However, this study does not aim to detect detailed geomorphic change, and we never claimed to be doing so. While having high-resolution Lidar or UAV-derived DEMs would be ideal, these data are currently unavailable and difficult to obtain, particularly for the vast area of the Upper Indus catchment that we are examining. A paper focused on detailed geomorphic change would be very interesting, but it would be a very different study than the one we have done.

**In line no. 98-113 of section 1. Introduction, we clarify that we are not looking at detailed geomorphic change, but rather the large-scale patterns of geomorphic change.**

**Please refer to Page No: 5 & Line No: 98-113**

"In the present study, we evaluated the **pattern of** spatial distribution of channel changes in the mountainous portion of the Upper Indus catchment due to the extreme precipitation event in the months of July and August 2022. We employed a channel slope-discharge product along the trunk channel of the Upper Indus River to estimate the anomalies in the stream power resulting from the anomalous precipitation event during July and August 2022. We used a random-forest-based machine learning approach to compare the observed and predicted intensity of precipitation and runoff by assessing the mean climatology of independent hydro-climatic variables. We further quantified the causal relationship between hydro-climatic drivers using nonlinear time series data over the event duration. We investigated the channel response of this episodic flood event by using the NDWI and EVI as change indicator metrics and comparing that to event characteristics such as anomalous precipitation, stream power, and channel metrics. We want to better understand the **general** controls on where and when these types of extreme hydrological events will substantially modify rivers and landscapes so associated geomorphic hazards can be better anticipated, and we also want to better constrain the potential role of these episodic events in driving long-term geomorphic change across the western syntaxial region".

**Comment 2.** Inadequate pre- and post-flood comparisons: The authors attempt to use MODIS-derived indices (NDWI, NDSI, EVI) for change detection (Lines 158-160). However, their approach has several limitations: (a) Temporal resolution: The authors don't specify the exact dates of the pre- and post-flood images used. Given that MODIS provides daily or 8-day composite products, the selection of these dates is crucial and could significantly affect the results (b) Spatial resolution: MODIS data typically has a spatial resolution of 250-1000m, which may be too coarse to capture detailed geomorphic changes in complex mountainous terrain. (c) Lack of quantitative analysis: The authors present qualitative descriptions of changes in these indices (Lines 350-356) but fail to provide a rigorous statistical analysis of the changes. For example, they could have conducted a pixel-by-pixel comparison and presented statistics on the percentage of areas showing significant changes. (d) Limited interpretation: While changes in vegetation indices can indicate flood impacts, the authors don't adequately address how these spectral changes relate to specific geomorphic processes or landforms. They make broad inferences about channel morphology changes (Lines 357-360) without directly linking spectral changes to field-observed geomorphic features. (e) Absence of complementary data: The use of optical indices alone is insufficient for comprehensive flood impact assessment. The authors could have strengthened their analysis by incorporating other remote sensing data, such as SAR imagery for flood extent mapping or high-resolution optical imagery for detailed change detection. The authors rely heavily on remote sensing indices (e.g., EVI, NDWI) to infer geomorphic changes (Lines 354-363). However, they provide no ground-truthing or field validation of these inferred changes. Without this validation, the reliability of their interpretations is questionable. The SAR/Landsat data can serve as a proxy in most cases.

**Authors Response:**

a) We compute the MODIS-based indices (NDSI, NDWI, and EVI) using the monthly mean composite of the daily datasets, maintaining a spatial resolution of 500 m. **This was added to the method in Section 3.1.**

**Please refer to Page No: 7 & Line No: 164-169**

"We used remote sensing-based indices to detect signatures of anomalous changes over the landscape. We computed these metrics over the monthly mean for July and August 2022,

using daily datasets of the MODIS-based normalized difference water index (NDWI), the normalized difference snow index (NDSI), snow albedo, EVI, and surface reflectance bands b1 and b2, which have a 500-meter spatial resolution".

b) Yes, it's true that the spatial resolution is not sufficient to capture detailed geomorphic change, but as we said above, the goal of this study is to look at regional and large-scale landscape patterns rather than detailed changes.

c) We didn't think that a pixel-by-pixel analysis was the appropriate way to check the robustness of the observed change. Instead, we accounted for the inherent variability in the metric by comparing the relative changes in 2022 to the changes in these metrics observed from 2002 to 2021.

d) This is a good point. We have compared the MODIS-derived change to both before and after optical imagery to ascertain the measurement. **We have included an example of this in the revised supplementary information (Figure S3).**

e) We attempted to use SAR data to look at flood inundation mapping; however, SAR proved unsuitable in this region because of the high relief and steep gradient valleys. One of the big advantages of MODIS data over optical imagery such as Landsat and Sentinel is that MODIS provides daily data as opposed to return periods of ~2 weeks.

**Comment 3.** Problematic causal analysis: The authors employ the PCMCI (Peter and Clark Momentary Conditional Independence) algorithm for causal discovery among hydro-climatic variables (Lines 244-270). While this is an advanced method for time series analysis, its application in this study has several significant issues: (a) Assumption of causal sufficiency: The PCMCI method assumes that all relevant variables are included in the analysis. However, the authors don't justify their selection of variables (temperature gradient, rainfall gradient, and anomalous change indicators) as a comprehensive set for describing the complex geomorphic system. Important factors like soil moisture, vegetation cover, or tectonic uplift rates are not considered, potentially leading to spurious causal relationships. (b) Linear relationships assumption: The authors use the ParCorr linear independence test (Line 261), which assumes linear relationships between variables. However, geomorphic and hydrological processes often exhibit

non-linear behaviors. The authors don't address this limitation or justify why a linear approach is appropriate for their data. (c) Temporal resolution mismatch: The authors use a maximum 2-day lag period (τmax = 2) for their analysis (Lines 267-268). This short-term focus may miss important longer-term causal relationships in the geomorphic system, which can operate on much longer timescales. d) Lack of robustness testing: The authors don't present any sensitivity analysis or robustness checks for their causal discovery results. It's crucial to test how the identified causal relationships change with different parameter choices (e.g., varying τmax or significance levels). e) Interpretation issues: The authors present their causal graph (Fig. 10) without adequately explaining how to interpret the results in the context of geomorphic processes. They don't link the statistical relationships found to physical mechanisms of landscape change. f) Temporal scope limitation: The analysis is limited to the July 1 to August 31, 2022 period (Lines 265-266). This narrow timeframe may not capture the full range of causal relationships relevant to the flood event, especially considering potential antecedent conditions or delayed effects.

**Author's Response:** There are a number of points raised about the causal analysis; several of these seem to stem from some confusion about what the causal analysis includes. We did this analysis to explore the hydro-climatic conditions of the high mountain flood. We are not attempting to look at the causality of long-term change or the full impact of the flood on the geomorphic system. The focus is on understanding the event's meteorological drivers and distribution of short-term impacts. **We have clarified this in Section 3.5 of the revised manuscript.**

**Please refer to Page No: 11 & Line No: 287-298**

"In order to evaluate the meteorological disturbances associated with the Upper Indus Flood of 2022, we identified the causal lag-connection between hydroclimatic variables, with a specific focus on exploring the meteorological conditions leading up to and during the flood event. We focused on identifying the short-term meteorological drivers that triggered the anomalous precipitation-driven high elevation flood and understanding the distribution of its immediate impacts within the Upper Indus catchment. We emphasize that this study does not attempt to explore the causality of long-term climatic changes or assess the full geomorphic consequences of the flood on the landscape. We deliberately limit the scope to comprehend the meteorological conditions and their direct impact on the flood in the July-August 2022 period.

By narrowing our focus to the short-term hydro-climatic interactions, we aim to offer insights into the key atmospheric processes and their role in shaping the event's severity rather than its broader or longer-term geomorphic impacts".

a)  We are using the causal analysis to investigate the meteorological drivers of the event and looking at temporal relationships and causality over two days. Factors that are constant on these time scales, such as uplift rates and distribution of vegetation cover are therefore not considered. The Upper Indus catchment's mean elevation is ~4000 m, with a predominantly glacial-fluvial regime. Much of the region is characterized by elevated low relief, steep landscapes, and a rain shadow effect, which collectively minimize the influence of soil moisture on the geomorphic system. Moreover, our preliminary analysis of soil moisture for this event did not yield significant results, further justifying its exclusion from the final model.

b)  It is true that geomorphic processes can exhibit non-linearity; however, our causal analysis is focused on the drivers of the flood and not the impacts and therefore does not attempt to describe any geomorphic processes. Non-linear methods might be able to capture more complexity, but our goal was to find the most direct and understandable causal pathways that led to the meteorological anomaly, drove this high-mountain flood, and significantly changed the landscape. In this context, the ParCorr test allowed us to effectively focus on these pathways. We acknowledge the potential for non-linearities and plan to explore them in future studies, but for this analysis, the linear approach provided clear and actionable insights into the causal relationships among the selected variables.

c)  As we stated above, we are not looking at long-term geomorphic processes, or attempting to do comprehensive geomorphic analysis, so the timescale is appropriate for the meteorological and hydrological variables that we are considering.

d)  It is important to clarify that the primary objective of our causal analysis was to identify the physical drivers of the extreme flood event across the Upper Indus catchment rather than directly model the geomorphic system itself. Our focus was on understanding how hydro-climatological variables, such as temperature and rainfall gradients, contributed

to the occurrence and magnitude of this high-mountain flood, which subsequently drove significant landscape changes. The causal graph provides insights into the temporal-lag relationships between these variables, highlighting the causal interrelationship between significant variables that triggered the flood. Although these statistical relationships don't aim to depict direct geomorphic processes, they aid in tracing the origins of the flood event, which subsequently influenced the geomorphic process. By identifying the flood's drivers, we can better understand the subsequent landscape changes, making this analysis a critical step in linking hydro-climatological dynamics to geomorphic outcomes in the context of high-elevation landscapes such as the Upper Indus catchment.

e)    Again, we are focused on the hydro-climatic drivers rather than the geomorphic processes. The EVI change can symbolize various processes such as inundation, erosion, and landsliding, but fundamentally, it signifies the region where a geomorphic or hydrologic process has eliminated the vegetation. **We added to the discussion of the EVI results by clarifying what the EVI change represents.**

**Please refer to Page No: 26-27 & Line No: 576-591**

"This study used the EVI change analysis as a significant event characteristic to capture the changes in the channel morphology triggered by the 2022 Upper Indus flood. The anomalous runoff events during the flood significantly altered channel geometry, and these changes were reflected in the spatial and temporal variations of EVI (Fig. 6). Geomorphic processes such as inundation, erosion, and landsliding have submerged or removed vegetation in areas marked by drastic shifts in EVI ranges (Anderson and Goulden, 2011). The reduction in EVI ranges along the steep channels highlights the expansion of water bodies during flooding, while the surrounding areas experienced erosion and landslides due to the extreme discharge. The broader geomorphic consequences of extreme hydrological events, such as river channel widening, sediment deposition, and riverbank erosion, frequently link to these changes in vegetation cover (Olen et al., 2016; Starke et al., 2020; Clift and Jonell, 2021; Scheip and Wegmann, 2021). While EVI cannot directly measure hydrologic parameters, its ability to reflect the loss of vegetation makes it a useful

proxy for assessing the intensity of geomorphic processes during floods. This capability is particularly important in high-mountain landscapes such as the Upper Indus, where steep landscapes and glacial fluvial regimes amplify the effects of extreme events".

    f)   It is true that we are not capturing the long-term geomorphic relationships, but again, this is not the focus of the causal analysis.

**Comment 4.** Unsupported stream power calculations: The author's novel approach to calculating stream power (Lines 204-224) incorporates precipitation data, but they fail to validate this method against established stream power calculation techniques or field measurements of actual stream power during the flood event.

**Authors Response:** This approach is not completely novel, and we have provided references in lines 214–215 for the method. Established stream power calculations assume uniform precipitation over the catchment, which is an assumption that was clearly violated in the Upper Indus catchment, which had a strong precipitation gradient. Therefore, we don't expect a match between typical stream power calculations and our precipitation-dependent stream power. To compare it with the actual streampower, we calculate the monthly mean streampower using the monthly climatology from the past 40 years. We have not attempted a full hydrological routing and modeling of the flood, nor have we attempted to reconstruct the flood discharges. Therefore, we do not anticipate a correlation between the calculated streampower anomaly and specific measurements of streampower during the flood. Furthermore, no discharge data has been made available from this region.

**Comment 5.** Insufficient error analysis and uncertainty quantification: The authors fail to adequately address uncertainties in their analysis, particularly in their Random Forest modeling (Lines 227-243). Specific issues include: a) Model performance metrics: No information is provided on standard evaluation metrics such as R-squared, RMSE, or Mean Absolute Error for the Random Forest predictions. b) Validation strategy: The authors don't specify their model validation approach (e.g., k-fold cross-validation, hold-out validation set). c) Feature importance: While they mention variable importance (Lines 372-380), they don't provide quantitative measures or visualizations of feature importance. d) Sensitivity analysis: There's no exploration of how model results change with different parameter settings or input variables. e) Propagation of

uncertainties: The authors don't discuss how uncertainties in their Random Forest predictions might affect subsequent analyses, such as the causal discovery or stream power calculations.

**Author's Response: We appreciate the opportunity to clarify our methodology and provide additional insights into our analysis.**

a) We acknowledge the importance of including standard evaluation metrics such as R-squared, RMSE, and mean absolute error (MAE) to assess the performance of our Random Forest model. We have already estimated these metrics and observed that the model performs well for the Upper Indus Catchment, followed by some upstream sub-catchments such as Shyok, Shinger, and Hunza. **In our revised manuscript, we have incorporated these metrics in the supplementary information (Table 1 & 2) to provide a comprehensive understanding of the model's predictive accuracy.**

b) The model validation strategy is indeed critical for ensuring the reliability of the predictions. In our original manuscript, we employed a time series cross-validation approach, which we did not adequately detail. We apologize for this oversight, and **we have provided a thorough explanation of our time series cross-validation methodology in the revised manuscript. Specifically, we discuss the choice of k, the process for splitting the data, and how this strategy helps mitigate overfitting while providing a robust estimate of model performance.**

**Please refer to Page No: 10 & Line No: 244-268**

"Based on the mean climatology of the last 40 years, we predict the daily anomalous precipitation and runoff intensity for the 2022 event and compare them with the observed actual values. We employed the highest significance variables, as well as precipitation and runoff data from 1982 to 2021, as a training set. We utilized a time series cross-validation approach in this study to evaluate the Random Forest model's performance in predicting precipitation and runoff during the Upper Indus catchment's high-elevation flood event in July and August 2022. Given the temporal dependency and sequential pattern of hydro-climatic data, using a normal K-fold cross-validation method could result in data leakage by allowing future data to inform past projections. To address this issue, we employed time

series cross-validation to maintain the data within chronological order. We trained the model using a sliding window method, gradually moving the training window forward in time with each iteration. Specifically, we designed the first training window to contain data from the first 30 years, leaving the last 10 years for testing. In each successive iteration, we increased the training window by one year and retrained the model on the expanded training set. We trained these models on meteorological variables obtained from the classification of the most significant, as well as other physical drivers associated with high-elevation flood episodes in the region. We evaluated the model's performance based on the accuracy of precipitation and runoff predictions, using metrics such as mean absolute error (MAE), mean squared error (MSE), and root mean squared error (RMSE) (please refer to SI). We computed these metrics for each rolling window to gain insight into the model's performance across various time periods, especially in the lead-up to the 2022 flood event. To utilize the independent variables to estimate these event characteristics, we first classify the hydro-climatic variables based on their higher importance using the RF classification approach. Then, by using the RF multivariate regression approach, we select only those independent variables with the highest significance to estimate anomalous precipitation and runoff intensity during the event duration".

c) & (d) We agree that investigating how model results change with different parameter settings and input variables is critical for understanding our model's stability and reliability. We appreciate your suggestion to conduct a comprehensive sensitivity analysis. We acknowledge that such analyses can be beneficial in certain contexts. However, due to the limitations of the available time series datasets, we were unable to perform a detailed sensitivity analysis or robust uncertainty quantification within the scope of this study. Additionally, while these approaches can provide deeper insights in some studies, we believe that, given the specific objectives and data constraints of our research, they would not yield significant additional insights into the key drivers of high-mountain floods and landscape changes in the Upper Indus catchment area. The goal of the Random Forest was to highlight the anomalous nature of the 2022 extreme event, and not to obtain a detailed prediction of rainfall and runoff; given these aims, the sensitivity analysis is not of high importance.

(e) We don't use the results of the Random Forest modeling in any of our other analyses, so there is no propagation of uncertainties possible. Our Random Forest, causality, and stream power/geomorphic change analyses are all completely separate and not dependent on each other.

**Comment 6.** Inappropriate temporal scale: The focus on only July-August 2022 (Lines 148-151) is too narrow to capture the full geomorphic impact of the flood event. This timeframe doesn't account for potential delayed landscape responses or longer-term geomorphic adjustments.

**Authors Response:** This is true, but as we have stated above, the purpose of this study is an overview of the physical drivers and initial impacts of the event, and we do not attempt a comprehensive or long-term picture of the full geomorphic impacts of the event. Again, this would be very interesting, but it would be a different study than the one we have done.

**Comment 7.** Insufficient error analysis and uncertainty quantification: The authors fail to adequately address uncertainties in their analysis, particularly in their Random Forest modeling (Lines 227-243). Specific issues include: a) Model performance metrics: No information is provided on standard evaluation metrics such as R-squared, RMSE, or Mean Absolute Error for the Random Forest predictions. b) Validation strategy: The authors don't specify their model validation approach (e.g., k-fold cross-validation, hold-out validation set).c) Feature importance: While they mention variable importance (Lines 372-380), they don't provide quantitative measures or visualizations of feature importance. d) Sensitivity analysis: There's no exploration of how model results change with different parameter settings or input variables. e) Propagation of uncertainties: The authors don't discuss how uncertainties in their Random Forest predictions might affect subsequent analyses, such as the causal discovery or stream power calculations.

**Authors Response**: **This comment is the same as Comment 5. Please refer to Author Response 5.**

**Comment 8.** Lines 331-363: The discussion of spatial distribution of hydro-climatic anomalies could be strengthened by including a statistical analysis of the relationships between different variables. Given the spatial nature of the remote sensing data, the authors should consider employing spatial statistical methods such as:

Geographically Weighted Regression (GWR): This technique could be used to explore the spatially varying relationships between precipitation anomalies and other variables like temperature, snowmelt, and runoff. GWR would allow the authors to identify how these relationships change across the study area, potentially revealing important local variations in flood response.

**Author Response:** Yes, this is a good point. We initially conducted this analysis and performed spatial interpolation using various hydro-climatic variables to capture the spatial distribution across the Upper Indus catchment. However, the significance of these observations during the monsoon period of 2022 was limited. Because this did not yield any insights, we did not include it in the manuscript. **We added this spatial analysis to the supplementary information (Figure S4).**

---

## Author Response (AR2)

1 **Anonymous referee #2**

2 **Comment:** The revised manuscript addresses several initial concerns, particularly in clarifying

3 the study's scope and clarity with methodological decisions.

4 Acknowledgment of Limitations: The absence of field-based validation should be clearly stated

5 as a limitation. This would lend transparency to the results while aligning expectations about

6 the study's scope. This could be framed as limitations more explicitly in the discussion.

7 **Author's Response:** Thank you for providing this valuable feedback on our manuscript. We

8 added the limitations of this study's scope in a section of Discussion 5.1.

9 **Please refer to Page No: 25 & Line No: 505-511**

10 "The present study of estimating stream power driven by anomalous precipitation identifies

11 regions across the Upper Indus catchment with high erosive potential during the 2022 flood

12 event. However, field validation or high-resolution pre- and post-event DEMs are required to

13 quantify the rate of erosion and patterns triggered by this extreme flood event. Our analysis

14 primarily draws upon pre- and post-observations from remote sensing-based indices, and

15 topographic analysis to spatially correlate estimated stream power with channel metrics".